# Differentiation of Isomeric TAT1-CARNOSINE Peptides by Energy-Resolved Mass Spectrometry and Principal Component Analysis

**DOI:** 10.3390/molecules30040853

**Published:** 2025-02-12

**Authors:** Alicia Maroto, Olivier Briand, Alessia Distefano, Filiz Arioz, Olivier Monasson, Elisa Peroni, Giuseppe Grasso, Christine Enjalbal, Antony Memboeuf

**Affiliations:** 1Univ Brest, CEMCA, CNRS, UMR 6521, 29238 Brest, France; alicia.maroto@univ-brest.fr (A.M.); distefano-alessia@libero.it (A.D.); filiz.arioz@univ-brest.fr (F.A.); 2Chemical Sciences Department, University of Catania, 95125 Catania, Italy; grassog@unict.it; 3CY Cergy Paris Université, CNRS, BioCIS, 95000 Cergy Pontoise, France; olivier.monasson@cyu.fr (O.M.); elisa.peroni@cyu.fr (E.P.); 4Université Paris-Saclay, CNRS, BioCIS, 92290 Orsay, France; 5Univ Montpellier, CNRS, ENSCM, IBMM, 34093 Montpellier, France; christine.enjalbal@umontpellier.fr

**Keywords:** isomeric peptides, carnosine, TAT1, high-resolution mass spectrometry, Principal Component Analysis, MS/MS, energy-resolved mass spectrometry

## Abstract

L-carnosine (Car) is an endogenous dipeptide with significant potential in drug discovery for neurodegenerative diseases, while TAT1, a small arginine-rich peptide derived from the HIV-1 trans-activator protein (TAT), is known to stimulate proteasome activity. In this study, three isomeric peptides were synthesised by incorporating the Car moiety at the N-terminus, C-terminus, or central position of the TAT1 sequence. To differentiate these isomers, high-resolution and energy-resolved CID MS/MS experiments were conducted. The resulting MS/MS spectra showed a high degree of similarity among the peptides, predominantly characterised by fragment ion peaks arising from arginine-specific neutral losses. Energetic analysis was similarly inconclusive in resolving the isomers. However, Principal Component Analysis (PCA) enabled clear differentiation of the three peptides by considering the entire MS/MS spectra rather than focusing solely on precursor ion intensities or major fragment peaks. PCA loadings revealed distinct fragment ions for each peptide, albeit with lower intensities, providing insights into consecutive fragmentation patterns. Some of these specific peaks could also be attributed to scrambling during fragmentation. These results demonstrate the potential of PCA as a simple chemometric tool for semi-automated peak identification in complex MS/MS spectra.

## 1. Introduction

Carnosine is a dipeptide found in high concentrations in muscle and brain tissues, where it plays a critical role in numerous biological processes [1,2,3]. Known for its strong antioxidant properties, carnosine protects cells from oxidative damage and glycation, both of which are implicated in ageing and chronic diseases [4,5,6,7,8,9]. It also aids glucose metabolism, with potential benefits in managing type 2 diabetes [10], and supports cognitive health by shielding neurons from oxidative stress [3,5,11,12]. In addition, carnosine enhances cellular energy metabolism [1,13], modulates nitric oxide (NO) pathways [14], and acts as a chelator of metals [15,16]. These diverse properties make carnosine a valuable candidate for various biomedical applications [1,17,18].

However, carnosine’s clinical use is constrained by its rapid enzymatic degradation [13,19,20]. In human serum, the enzyme carnosinase hydrolyses carnosine into beta-alanine and histidine, significantly lowering its bioavailability and therapeutic efficacy. To overcome this limitation, researchers have investigated structural modifications to carnosine to enhance its stability and resistance to enzymatic breakdown. One promising strategy involves conjugating carnosine with organic molecules, producing derivatives or analogues that preserve its biological functions while being less susceptible to carnosinase degradation [13,19,20].

In this perspective, one possible strategy is represented by conjugation with peptide chains with high stability and resistance to proteolytic degradation, such as TAT1, an arginine-rich cell-penetrating peptide (CPP) derived from the transactivator of transcription (TAT) protein of HIV-1, which is known for its ability to efficiently traverse cellular membranes [21,22,23]. This property facilitates the intracellular delivery of diverse therapeutic molecules, ranging from small compounds to nucleic acids, proteins, and even liposomes, regardless of their size or physicochemical characteristics. Owing to this versatility, TAT1 has become a valuable tool in drug delivery and biomedical research.

Therefore, the combination of TAT1 with carnosine presents an intriguing possibility. While carnosine’s biological functions, such as its antioxidant properties and metabolic regulation, are well-documented, its therapeutic application is limited by low bioavailability due to enzymatic degradation. TAT1’s ability to enhance cellular uptake could potentially improve carnosine delivery, mitigating this limitation and expanding its applicability [24,25,26].

In this context, the TAT1 sequence was modified to incorporate carnosine at three distinct positions, resulting in three isomeric peptides [24,25]. In the first sequence, Car-TAT1, carnosine is attached at the N-terminal position. In the second, T-Car-T, carnosine is inserted in the middle of the sequence. Finally, in TAT1-Car, carnosine is positioned at the C-terminal end. The original TAT1 peptide and carnosine, along with the three modified isomeric peptides, are presented in Figure 1.

These TAT1-carnosine peptides were successfully characterised using quantum dots [24,26]. The fluorescent properties of the quantum dots were influenced by the position of carnosine within the peptide sequence, demonstrating their potential application for the quantification of mixtures of isomeric peptides. Additionally, Time-of-Flight Secondary Ion Mass Spectrometry (ToF-SIMS) was employed to differentiate the three isomeric TAT1-carnosine peptides [25]. However, under the high-energy conditions of ToF-SIMS, specific structural information was largely lost due to non-specific fragmentation. This resulted in fragment ions predominantly associated with individual amino acids rather than sequence-specific peptides. Despite this limitation, Partial Least Squares Discriminant Analysis (PLS-DA) successfully differentiated the three peptides. Variations in fragmentation patterns, particularly in fragment ion abundances, facilitated the discrimination of these isomeric peptides [25].

Tandem mass spectrometry (MS/MS) is a powerful tool for distinguishing between isomers, provided that each isomer generates fragment ions specific to its structure [27,28]. However, in certain cases, isomers cannot be differentiated through straightforward visual inspection of their MS/MS spectra. Consequently, alternative MS strategies have been developed to address this challenge, including the kinetic method [29,30], Ion Mobility Mass Spectrometry (IM-MS) [31,32,33,34,35], and energy-resolved mass spectrometry (ER MS) [30,36,37,38,39,40,41,42,43,44,45,46,47].

In ER MS, collision-induced dissociation (CID) experiments are conducted at multiple excitation voltages to generate MS/MS spectra that reveal differences in fragmentation behaviour. Bartolucci et al. successfully utilised ER MS data with multilinear regression models to quantify co-eluted isomers [41,42,44,45,46,47,48]. Additionally, Memboeuf et al. demonstrated that ER MS, particularly through Survival Yield (SY) plots, can be leveraged for structural analysis of isobaric compounds [39,49], quantification of isomeric and isobaric mixtures [49,50,51,52,53], and the elimination of isobaric interferences in LC–MS [40,54].

The objective of this study is to evaluate whether the position of carnosine within the peptide sequence can be differentiated using CID tandem mass spectrometry, eventually using the ER MS strategy. First, the MS/MS spectra obtained in high resolution were analysed visually, followed by ER MS analysis using SY plots and the breakdown curves of the predominant fragment ions. Subsequently, principal component analysis (PCA) was employed as a chemometric tool to differentiate the three peptides and identify specific fragment ions associated with the position of carnosine. Finally, MS/MS spectra are discussed in light of PCA analysis, including both conventional fragmentation [55] and scrambling [56].

## 2. Results and Discussion

### 2.1. Energy-Resolved Mass Spectrometry and Breakdown Curves of TAT1-Carnosine Peptides

High-resolution ER MS experiments were conducted by measuring the MS/MS spectra of doubly protonated TAT1-carnosine peptides. The precursor ion selected for fragmentation was the doubly protonated adduct [M + 2H]^2^⁺, with *m*/*z* = 895.04. MS/MS spectra were acquired at acceleration voltages ranging from 20 V to 42 V.

MS/MS spectra of the three peptides exhibit a high degree of similarity. The precursor ion peak remains the most abundant across all three peptides, even at high excitation voltages. Appendix A show the high-resolution MS/MS spectra of the three isomeric TAT1-carnosine peptides at 36 V. The most abundant peaks, located in the *m*/*z* 800–900 range, are common to all three isomers and include ions at *m*/*z* 886.52, 874.03, 865.51, and 857.00.

Figure 2 illustrates the high-resolution MS/MS spectrum of Car-TAT1, focusing on the *m*/*z* range of 800–900. The same fragment ions were also observed for TAT1-Car and T-Car-T. Peaks highlighted in red correspond to the conventional [b_14_]^2+^ ion and its consecutive neutral losses, primarily associated with the presence of arginine. The [b_14_]^2+^ ion, at *m*/*z* 877.52, corresponds to the loss of water and the formation of either a protonated oxazolone on the last residue at the C-terminal position or a macrocycle [57,58,59]. The other fragments (in black) do not correspond to conventional fragments and are uniquely due to neutral losses from arginine. These involve molecules such as ammonia (NH_3_), carbodiimide (CH_2_N_2_), and guanidine (CH_5_N_3_). Since the peptides are doubly charged, the masses of these neutral losses are halved. For example, the loss of ammonia corresponds to Δ*m*/*z* = −17.0265/2 ≈ −8.51, the loss of water to Δ*m*/*z* = −18.0105/2 ≈ −9.01, the loss of carbodiimide to Δ*m*/*z* = −42.0218/2 ≈ −21.01, and the loss of guanidine to Δ*m*/*z* = −59.0483/2 ≈ −29.52.

Figure 2 also reveals that the 0.5 *m*/*z* difference between certain peaks corresponds to the loss of H_2_O rather than NH_3_, further confirming that these are doubly charged ions. Carbodiimides and guanidines are derived from the termini of arginine side chains. The spectra indicate that fragmentation involves consecutive neutral losses, meaning that these molecules are lost multiple times, resulting in the observation of a large number of fragment ions across a wide spectral range. However, the detected ions provide no structural information, appear virtually identical, and are highly intense for each peptide. Consequently, they do not allow for differentiation among the peptides. Since the three peptides are isomers, the inability to distinguish them highlights the challenge in resolving isomeric structures based solely on the visual inspection of these MS/MS spectra.

To differentiate the three isomeric peptides, we employed energy-resolved mass spectrometry (ER MS), focusing specifically on Survival Yield (SY) curves and breakdown collision curves of the most abundant fragment ions.

The Survival Yield (SY) was calculated at each excitation voltage as the ratio of the precursor ions peak intensity to the Total Ion Current (TIC) [39,49,50,51,52,53]:(1)$$SY=\frac{I_{\mathrm{precursor}}}{I_{\mathrm{precursor}}+\sum I_{\mathrm{fragment}}}$$
where *I*_precursor_ is the intensity of the precursor ions peak, and *I*_fragment_ is the intensity of each fragment ions peak obtained from the MS/MS experiment. SY curves were generated by plotting SY values against the acceleration voltage (see Figure 3). The resulting curves show a high degree of overlap, making it impossible to differentiate between TAT1-Car and T-Car-T. However, Car-TAT1 exhibits a slightly lower SY across the voltage range, allowing for a subtle distinction from the other two peptides.

To further distinguish between the three isomeric peptides, breakdown curves of the major fragment ions were plotted for *m*/*z* 886.52, 874.03, 865.51, and 857.00 (Appendix A). The abundance of each fragment ion was calculated similarly to the SY, as the intensity of the fragment ions peak was divided by TIC and plotted against the acceleration voltage to generate the breakdown curve. In this case, T-Car-T exhibited lower fragmentation of the major fragment ions, which aligns with its slightly higher SY compared to the other two peptides. Among the fragment ions, *m*/*z* 857 shows the greatest differentiation between the three peptides. However, this fragment ion is not selective and cannot be uniquely attributed to any specific peptide. To enhance differentiation and identify specific fragment ions, Principal Component Analysis (PCA) was applied, providing additional insights into the structural distinctions among the three peptides.

### 2.2. Principal Component Analysis for the Differentiation of TAT1-Carnosine Peptides

Principal Component Analysis (PCA) is a powerful statistical technique used to reduce the dimensionality of complex datasets while preserving the most relevant information. By transforming the original variables into a set of orthogonal components, PCA identifies patterns and highlights subtle differences within the data [60]. In this study, the variables correspond to the *m*/*z* values from the MS/MS spectra, and PCA facilitates the differentiation of the three isomeric peptides.

The scores represent the projection of the peptides onto the principal components, revealing clustering or separation in the data. The loadings indicate the contribution of each variable (*m*/*z*) to the principal components, providing insight into which peaks are most responsible for differentiation.

PCA was applied to the full MS/MS spectra acquired at 36 V, covering the *m*/*z* range from 100 to 900. At this voltage, the Survival Yield (SY) was approximately 0.2 (Figure 4), indicating a high degree of fragmentation and the potential to identify specific fragment ions beyond those resulting from neutral losses. Prior to PCA, each MS/MS spectrum was normalised to TIC and centred.

The results of the PCA are presented in Figure 4. The score plot (Figure 4a) shows that PC1, which explains 52.05% of the variance, separates T-Car-T from TAT1-Car and Car-TAT1. This differentiation in PC1 is primarily driven by peaks at *m*/*z* 886.53, 865.52, 874.03, and 857, as revealed in the loading plot (Figure 4b). These peaks exhibit positive loadings and are more intense in TAT1-Car and Car-TAT1 than in T-Car-T. However, as shown in Appendix A, the corresponding fragment ions, resulting from neutral losses, are not specific to any peptide.

In contrast, T-Car-T is characterised by peaks with strongly negative loadings in PC1, particularly the peak at *m*/*z* 843.02. This fragment ion is clearly associated with T-Car-T, as shown in the loading plot. The breakdown curve for this fragment ion (Figure 5) demonstrates its specificity to T-Car-T, as no intensity is observed for the MS/MS spectra of TAT1-Car or Car-TAT1. This absence underscores its utility as diagnostic fragment ions peak for T-Car-T.

PC2, which explains 47.95% of the variance, differentiates TAT1-Car from Car-TAT1, a distinction not achieved by PC1. The loadings plot highlights two key fragment ions contributing to this separation: *m*/*z* 826.51 for TAT1-Car and *m*/*z* 773.47 for Car-TAT1.

Notably, *m*/*z* 843.02, 826.51, and 773.47 were identified as specific to T-Car-T, TAT1-Car, and Car-TAT1, respectively, demonstrating that the combined observation of the score and loading plots reveals unique markers for each peptide. Figure 5 highlights these specific peaks, underscoring the utility of PCA in uncovering structural differences that are not easily identified through visual inspection of MS/MS spectra.

The higher loadings of PC1 are predominantly associated with neutral losses, which do not provide structural information. Analysis of the loadings line plot (Appendix A) indicates that the most significant contributions to the PCA occur in the *m*/*z* range of 600–900. However, the region between *m*/*z* 800 and 900 primarily reflects neutral losses, which are non-specific and lack structural relevance. To focus only on specific peaks that provide structural information, we also performed PCA on a reduced section of the MS/MS spectrum, specifically from *m*/*z* 600 to *m*/*z* 800. This approach excludes the influence of neutral losses and enhances the structural interpretability of the results.

The results of this focused PCA are presented in Figure 6. The most contributing peaks are concentrated at *m*/*z* values above 700 (see line plot in Appendix A). The combined view of the score and loading plots (Figure 6a,b) reveals distinct patterns. Some peaks are highly correlated and diagnostic of a specific peptide. For example, Car-TAT1 displays characteristic peaks that are highly correlated at *m*/*z* values 773.47, 764.95, 752.46, and 743.95. Similarly, T-Car-T exhibits correlated peaks at *m*/*z* values 772.98, 764.47, 751.88, and 743.46.

In contrast, peaks related to TAT1-Car are less correlated. The most representative peak for TAT1-Car is *m*/*z* 796.98, along with other less correlated fragment ions, such as *m*/*z* 788.47 and 775.98. Additionally, some peaks originate from at least two peptides, making them less selective. For instance, peaks at *m*/*z* 787.98, 779.47, or 782.52 are associated with more than one peptide.

Moving forward, we will investigate the peaks identified as characteristic of each peptide to better understand their potential for distinguishing between peptides.

### 2.3. Specific Fragment Ions for CAR-TAT1 Identified by PCA

The loading plot of PC1 vs. PC2 in Figure 6b highlights that certain fragment ions (*m*/*z* 793.95, 764.96, 752.46, 743.95, and 735.43) are strongly correlated. This is evident from the alignment of their loading vectors, which form small angles with one another, resulting in a correlation coefficient close to 1. Although the magnitudes of their loadings vary, their similar vector directions underscore the high correlation between these fragment ions.

Figure 7 highlights the peaks with the highest loadings associated with Car-TAT1. These loadings represent three highly correlated peaks: *m*/*z* 764.96; 752.46; and 743.95. Interestingly, all these peaks are specific to Car-TAT1, demonstrating that the information contained in the loadings allows for the identification of specific peaks.

Furthermore, the loadings provide additional valuable information: their magnitudes reflect the abundance of the fragment ions. Higher loading magnitudes correspond to greater fragment ion abundances, enhancing the sensitivity for detecting Car-TAT1. This relationship is confirmed in the breakdown curves shown in Figure 7. The highest loading was observed for *m*/*z* 764.96, which also exhibited the greatest abundance, whereas *m*/*z* 743.95 displayed a lower loading and a corresponding decrease in abundance in the breakdown curve.

The MS/MS spectrum shown in Figure 8 confirms that the correlation observed with the aligned loading vectors is logical, as all the identified fragment ions correspond to neutral losses from the same doubly charged ion [y_12_]^2+^. However, Figure 8 also shows that [y_12_]^2^⁺ is barely detected. This suggests that losses of several units of H_2_O and NH_3_ occur prior to the separation of carnosine and TAT1, which explains the low intensity observed for *m*/*z* 790.99.

Additionally, a peak at *m*/*z* 209.10 was identified as ion b_2_, associated with y_12_, likely due to a charge separation fragmentation process (see Appendix A). This observation further supports the cleavage site between the histidine of carnosine and the glycine of TAT1 (Appendix A).

Overall, the loadings demonstrate their ability to identify specific fragmentation patterns for Car-TAT1. Furthermore, as previously observed in the breakdown curves, the MS/MS spectrum (Figure 8) reaffirms that the loading magnitude is proportional to the fragment abundance. Specifically, *m*/*z* 773.47 has the highest loading and also the greatest abundance. This peak is followed by *m*/*z* 752.46 and 765.50, which show slightly lower abundances and are almost overlapping in the biplot. The peak *m*/*z* 743.95 exhibits a lower loading and correspondingly lower abundance, as shown in Figure 8. This trend is further confirmed with peaks *m*/*z* 735.43 and 756.45, which also display lower abundances and lower loadings. Figure 8 illustrates that all these peaks are formed through neutral losses from [y_12_]^2^⁺, showing that the correlation of the loadings is due to a consecutive fragmentation pattern.

### 2.4. Specific Fragment Ions for T-Car-T Identified by PCA

Figure 9 highlights the fragment ions with the highest loadings associated with T-Car-T: *m*/*z* 772.98; 764.47; and 751.98. These peaks are specific to T-Car-T, as they are not observed for Car-TAT1 or TAT1-Car. The loadings corresponding to these peaks are strongly correlated, as evidenced by the alignment of their loadings vector, indicating a correlation coefficient close to one (Figure 6b).

This observation suggests a consecutive fragmentation pattern similar to that previously noted for Car-TAT1. Specifically, from *m*/*z* 772.98, the neutral loss of ammonia results in the consecutive peak *m*/*z* 764.47, while the loss of guanidine from *m*/*z* 772.98 results in the peak *m*/*z* 751.98.

Furthermore, the breakdown curves of these two consecutive peaks (*m*/*z* 764.47 and 751.98) are shifted to higher energies and exhibit lower abundances, consistent with their formation from *m*/*z* 772.98. This decrease in intensity is also reflected in the loading magnitude: the loading for *m*/*z* 772.98 has a higher magnitude than those of its consecutive peaks, *m*/*z* 764.47 and 751.98. This pattern once again demonstrates the proportional relationship between abundance and loading magnitude.

The MS/MS spectrum shown in Figure 10 reveals that these fragment ions are generated primarily through scrambling [56], except for the fragment ion [RRRP]^+^, which is formed by cleavage at both ends of the peptide, resulting in an internal b ion. Fragmentation of peptides by scrambling refers to a process in which peptide ions undergo fragmentation through rearrangements. In scrambling, the peptide may fragment in a way that involves less predictable bond cleavages or rearrangements, resulting in fragment ions that differ from those produced by conventional peptide fragmentation. The change in the position of amino acids in the sequence during fragmentation leads to different combinations of amino acids in a fragment ion. Scrambling can complicate the interpretation of mass spectrometry data, as it introduces unexpected peaks.

For T-Car-T, due to scrambling, carnosine (Car) ends up in the C-terminal position and then separates from TAT1. Schematically, the following transformation occurs:GRKKRRQ-βAla-H-RRRPS → RRRPS-GRKKRRQ-βAla-H → RRRPS-GRKKRRQ

This final sequence would correspond to a b-type ion with *m*/*z* 781.99 (labelled b_β_^2^**⁺** in Figure 10 for clarity). However, this peak is not detected in the MS/MS spectrum. Instead, a peak at *m*/*z* 772.98 is observed, corresponding to the neutral loss of water from b_β_^2^⁺. This suggests that the neutral loss of water occurs prior to fragmentation by scrambling. Figure 10 shows several fragment ions generated by neutral losses of b_β_^2^⁺.

Also for T-Car-T, the loadings demonstrate their ability to identify specific fragmentation patterns. Moreover, it is important to note that these fragment ions would be quite difficult to identify without PCA because they are generated by scrambling, leading to unexpected peaks. Furthermore, Figure 10 reaffirms that the alignment of the loading vectors is logical, as all these peaks are generated by consecutive neutral losses from the same fragment ion generated by scrambling, b_β_^2^**⁺**. This further supports the idea that the correlation observed in the loadings can serve as an indicator of consecutive fragmentation patterns for T-Car-T.

Additionally, as for Car-TAT1, the loading magnitude is proportional to fragment ion abundance. Specifically, *m*/*z* 772.98 exhibits the highest loading and the highest abundance. This peak is followed in abundance by *m*/*z* 764.47, which shows a lower loading magnitude. This trend is further confirmed by *m*/*z* 751.98, 743.46, and 755.96, which display progressively lower abundances and lower loading magnitudes.

### 2.5. Specific Fragment Ions for TAT1-Car Identified by PCA

Figure 11 highlights some of the peaks identified as characteristic of TAT1-Car in the PCA (Figure 6b). Two peaks with the highest loading magnitudes were selected: *m*/*z* 796.98 and *m*/*z* 788.47. Their vector loadings indicate a significant correlation between the two peaks, even though the vectors are not perfectly aligned, suggesting that the correlation is strong but not complete. Two highly correlated additional peaks, *m*/*z* 723.40 and *m*/*z* 668.93, also correspond to loadings characteristic of TAT1-Car, but their magnitudes are lower.

Figure 11 reveals that *m*/*z* 796.98 is not unique to TAT1-Car, as T-Car-T also displays this peak, albeit at a much lower intensity. Nonetheless, this peak is identified as characteristic of the PCA due to its significantly higher abundance in TAT1-Car. All other peaks were specific to TAT1-Car, demonstrating that the loadings provide valuable information for identifying specific peaks of TAT1-Car.

As shown in Figure 11, the abundance of these peaks in TAT1-Car is proportional to their loading magnitudes. Specifically, the peak *m*/*z* 796.98 has a slightly higher abundance than *m*/*z* 788.47. Conversely, the peaks *m*/*z* 668.93 and *m*/*z* 723, which have nearly overlapping loadings, display similar abundances. In line with their lower loading magnitudes, these peaks exhibit much lower abundances compared to *m*/*z* 796.98 and *m*/*z* 788.47.

Figure 12 shows the MS/MS spectrum of TAT1-Car. In this case, the fragment ions of this peptide primarily result from conventional fragmentation and neutral losses, making them easier to identify than the fragment ions of T-Car-T. These conventional fragment ions consist mainly of b and y ions carrying one or two charges (see Appendix A). Additional fragment ions are attributed to neutral losses associated with the b or y fragment ions. For instance, *m*/*z* 668.93 is formed by the neutral loss of carbodiimide from the b_10_ fragment ion.

It should be noted that the formation of b_m_ – y_n-m_ ion pairs, which confirm cleavage sites, was observed only for b_4_– y_10_ and b_10_ – y_4_. Among the ions identified, two peaks correspond to neutral losses from the precursor ion: *m*/*z* 796.98 and its consecutive peak, *m*/*z* 775.98, generated by the neutral loss of carbodiimide. Therefore, it is logical that the peak *m*/*z* 796.98 is not entirely selective to TAT1-Car, as it arises solely from neutral losses of the precursor ion and does not carry structural information. In contrast, although *m*/*z* 775.98 is also formed from neutral losses of the precursor ion, this fragment ion was selective to TAT1-Car at 36 V. This explains the position of its loading vector, which is more correlated with the conventional fragment ions characteristic of TAT1-Car.

## 3. Materials and Methods

### 3.1. Chemicals and Synthesis of Peptides

Acetonitrile and formic acid (LC–MS grade) were obtained from Honeywell (Guyancourt, France). Milli-Q water was produced using a Milli-Q water purification system (Millipore, Bedford, MA, USA).

Peptides were synthesised by microwave-assisted solid-phase peptide synthesis using a Liberty Blue Microwave Automated Peptide Synthesiser (CEM Corporation, Matthews, NC, USA) following the standard protocol for Fmoc/tBu chemistry on a 0.1 mmol scale [24,25]. Fmoc-deprotection cycles consisted of 15 s at 75 °C (155 W) and 30 s at 90 °C (30 W). The synthesised peptides had the following sequences (see Figure 1):TAT1-Car: GRKKRRQRRRPS-βAla-H;Car-TAT1: βAla-HGRKKRRQRRRPS;T-Car-T: GRKKRRQ-βAla-H-RRRPS.

All peptides were isomers with the molecular formula C_72_H_133_N_37_O_17_ and a monoisotopic mass of 1788.0654 Da. Peptides were purified using semi-preparative RP-HPLC with a fully automated Waters Preparative System equipped with a Phenomenex Synergi 4µ Fusion-RP C18 column (150 × 10 mm) operated at a flow rate of 4 mL·min at 25 °C. The solvent systems used were A (0.1% trifluoroacetic acid in water) and B (0.1% trifluoroacetic acid in acetonitrile).

Peptide purity was analysed by analytical RP-UPLC ESI-MS using a Waters Acquity UPLC coupled to a Waters 3100 ESI-SQD MS equipped with a Luna Omega PS C18 column (1.6 µm, 2.1 × 50 mm) operated at 35 °C with a flow rate of 0.6 mL·min. The solvent systems were the same as those used for semi-preparative HPLC. All peptides were obtained with a purity of ≥90%. Peptide solutions were prepared at a concentration of 10 µmol L^−1^ in 70:30 acetonitrile/water containing 1%v formic acid.

For calibration in HR-MS, a solution of sodium formate (HCOONa) and a leucine enkephalin solution (100 pg µL^−1^) were used. The HCOONa solution was prepared by mixing 100 µL of 0.1 M NaOH with 200 µL of 10%v formic acid (Amresco Inc., Fontenay-sous-Bois, France), followed by the addition of 20 mL of 80:20 acetonitrile/water. The leucine enkephalin solution was prepared by diluting a 1 mg·mL^−1^ stock solution 1:10 in 0.1:50:50 formic acid/methanol/water. The stock solution was prepared using leucine enkephalin acetate salt hydrate (Sigma-Aldrich, Saint Louis, MO, USA, ≥95% HPLC grade).

### 3.2. Energy-Resolved Mass Spectrometry

High-resolution MS/MS spectra of the peptides were acquired using a Waters Q-ToF Synapt XS high-resolution mass spectrometer. Data acquisition and MS/MS processing were performed with MassLynx software 4.2 (Waters Corp., Milford, MA, USA).

The analyses were carried out by direct infusion at a flow rate of 10 µL·min⁻^1^. The precursor ion selected for MS/MS experiments was the doubly protonated peptide [M + 2H]^2^⁺ (*m*/*z* 895.03).

All mass measurements were performed in positive ion mode using an electrospray source (ESI+). The temperature of the source was set at 90 °C; nitrogen was used as the desolvation gas at a flow rate of 300 L·h⁻^1^ and a temperature of 250 °C. The cone gas flow was set at 50 L·h⁻^1^, and the nebuliser gas pressure was set at 4 bars. The capillary voltage was set at 3.8 kV. During the CID MS/MS experiments, the quadrupolar mass analyser was used with LM and HM resolutions set to 20 and 15, respectively. Argon was used as the collision gas with a pressure of 8.15 × 10⁻^3^ mbar inside the collision cell during MS/MS experiments (uncorrected gauge reading).

All MS/MS spectra were acquired in the spectral range from 50 to 2000 *m*/*z*, with each acquisition lasting 1 min. The instrument resolution was higher than 15,000. A solvent rinse (acetonitrile/water 7:3) was performed after each solution change to avoid cross-contamination.

The MS/MS spectra obtained were processed as follows: smoothed using a Savitzky–Golay algorithm with a 5-channel window and centred. The high-resolution MS/MS spectra were acquired at acceleration voltages ranging from 20 V to 42 V.

### 3.3. Multivariate Analysis

The multivariate analysis was carried out using R software (version 4.1.3) [61] and RStudio (version 2022.2.0.443) [62]. Principal component analysis (PCA) was performed with the package “mdatools” [63]. The MS/MS spectra were normalised to the total ion current (TIC), ensuring all MS/MS spectra had the same value range from 0 to 1. This range corresponds to the relative abundance of each peak. The normalised data were centred before performing PCA. PCA was subsequently conducted using the NIPALS (Nonlinear Estimation by Iterative Partial Least Squares) algorithm.

## 4. Conclusions

The high rigidity of the TAT1-carnosine peptides, attributed to the highly basic nature of these anomalously arginine-rich peptides, resulted in highly similar high-resolution MS/MS spectra for the three isomeric peptides. The predominant fragment ions, primarily observed in the 800–900 *m*/*z* range, were consistent across all three peptides. These fragment ions arose mainly from neutral losses of arginine side chains and lacked structural information regarding the position of the carnosine residue. Furthermore, energy-resolved mass spectrometry proved inconclusive for distinguishing between the three isomeric peptides. The SY curves were nearly superimposable, and the breakdown curves of the predominant fragment ions were non-specific and remarkably similar for all three peptides.

In contrast, principal component analysis (PCA) demonstrated its effectiveness in differentiating the three isomeric peptides by identifying key fragment ions across the full MS/MS spectra. Unlike approaches that focus solely on the precursor ion or the most intense fragment ions, PCA enabled clear discrimination of the peptides. The PCA loadings revealed the presence of characteristic fragment ions for each peptide, including those with lower intensities. These correlated loadings also provided insights into consecutive fragmentation patterns. Moreover, the magnitude of the PCA loadings was proportional to the abundance of the characteristic fragment ions, offering the possibility of identifying the most abundant specific fragment ions for a given peptide. Notably, PCA identified low-intensity fragment ions that were difficult to detect through visual inspection, particularly in the case of T-Car-T, where fragment ions generated by scrambling were unexpected and challenging to identify manually.

The specific fragment ions identified through PCA further highlight the influence of carnosine’s position within the peptide sequence on its fragmentation behaviour. For Carn-TAT, where carnosine is located at the N-terminal position, fragmentation primarily results in its dissociation from TAT, producing only the b_2_ and y_12_ conventional ions. In contrast, TAT1-Car, with carnosine at the C-terminal position, generates a broader range of conventional fragment ions. Interestingly, when carnosine is positioned in the middle of the sequence, as in T-Car-T, fragmentation involves scrambling, with carnosine rearranging to the C-terminal position before cleavage. This behaviour suggests that the C-terminal position of carnosine facilitates peptide fragmentation.

Looking ahead, multivariate calibration methods such as Partial Least Squares (PLS) could be explored to evaluate the potential of chemometric approaches for quantifying mixtures of the three carnosine-TAT1 peptides. Additionally, alternative MS fragmentation techniques, such as electron-based techniques (ExD), could be employed, as the TAT1 peptides easily generate multiple protonated signals due to their arginine-rich structures. This would help prevent arginine side chain losses, which lack structural information, and instead promote more backbone fragmentations that are more closely related to the peptide’s structure, which is suitable for the studied sequences [64,65,66,67]. Finally, ion-mobility mass spectrometry could be investigated to evaluate its ability to distinguish between the three isomeric peptides based on their gas-phase conformations, as the position of the carnosine dipeptide can induce different structural arrangements, leading to variations in their Collision Cross Section (CCS) and, thus, measured drift times, thereby enabling differentiation [68,69].

It may be worth highlighting that classical peptide sequencing methods using CID activation, employed in most MS/MS instruments and based on the mobile proton model, typically lead to backbone fragmentations through amide bond cleavage [70]. Ideally, this produces comprehensive and uniform fragmentation patterns, generating a broad series of complementary b and y ions. Machine learning-based classification methods have been applied in this context to assign their b and y nature, improving the accuracy and efficiency of de novo sequencing [59,71,72]. However, in the case of the studied peptides, charge sequestration occurs due to the presence of many arginine residues, which are the most basic amino acid residues in the gas phase. As a result, fragmentation deviates from the ideal scenario, leading to multiple uninformative neutral losses and parasitic fragmentation pathways, such as the formation of internal fragments and scrambled sequences. This significantly complicates spectral assignment [73,74]. Moreover, no reference spectral databases are available for comparison, as these compounds are not true peptides due to the presence of a beta amino acid. Consequently, despite extensive knowledge and methodologies for peptide sequencing using MS/MS, such complex and atypical cases require dedicated investigations. This highlights the need for complementary and orthogonal approaches, such as the PCA, for identifying non-conventional fragments, as demonstrated in the present study.

## Figures and Tables

**Figure 1 molecules-30-00853-f001:**
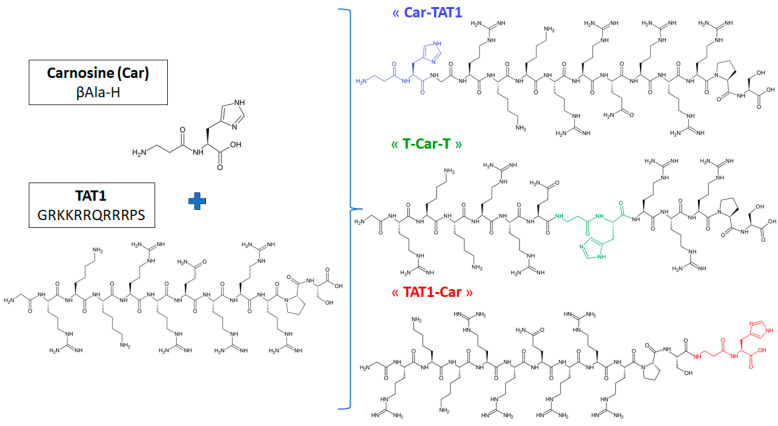
Amino acid sequences of carnosine (Car) and TAT1, along with the three isomeric peptides studied, are shown. The colour of carnosine indicates its position within the peptide: in Car-TAT1, carnosine (blue) is bound to the N-terminus; in TAT1-Car, carnosine (green) is bound to the C-terminus; and in T-Car-T, carnosine (red) is located in the middle of the TAT1 amino acid sequence.

**Figure 2 molecules-30-00853-f002:**
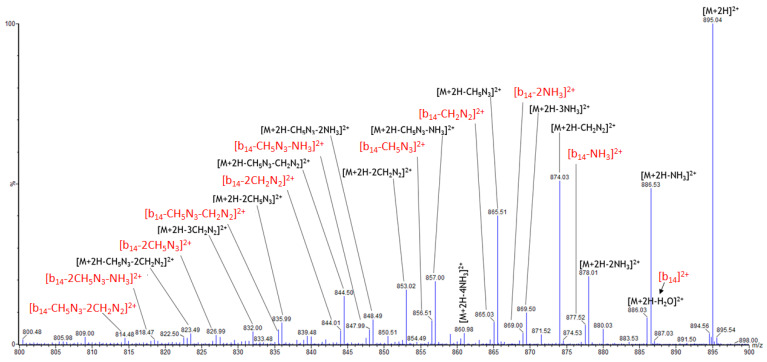
High-resolution MS/MS spectrum of Car-TAT1 from 800 to 900 *m*/*z*. Acceleration voltage: 36 V. The fragment ions are due to consecutive neutral losses, and they do not provide information on the position of carnosine. This part of the MS/MS spectrum is similar to the three isomeric peptides. The b-type conventional fragment and its consecutive neutral losses are highlighted in red, while the remaining fragments (in black) correspond to non-conventional fragments related to the neutral losses of arginine.

**Figure 3 molecules-30-00853-f003:**
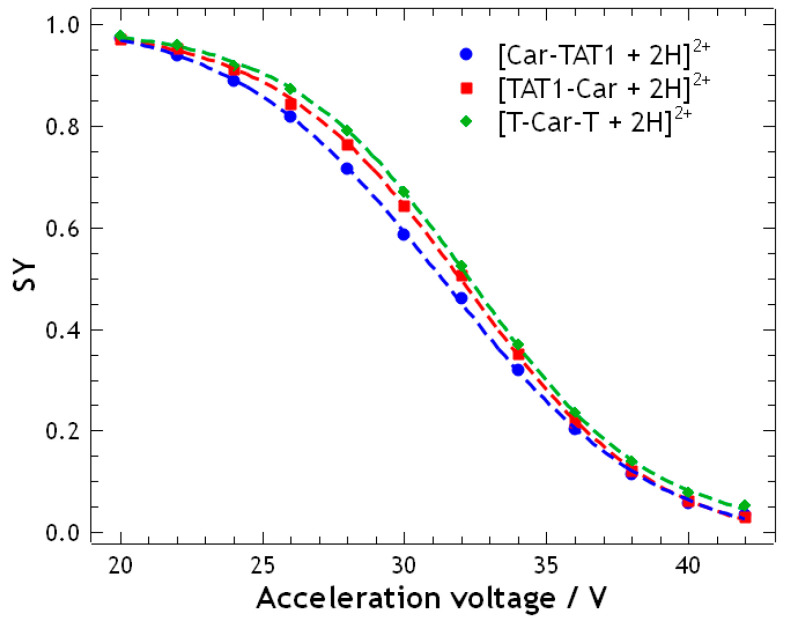
Survival Yield (SY) curve of the double protonated isomeric peptides: Car-TAT1 (blue); TAT1-Car (red); and T-Car-T (green). SY was calculated from high-resolution MS/MS spectra performed at acceleration voltages ranging from 20 V to 42 V. The SY curves are similar for the three peptides.

**Figure 4 molecules-30-00853-f004:**
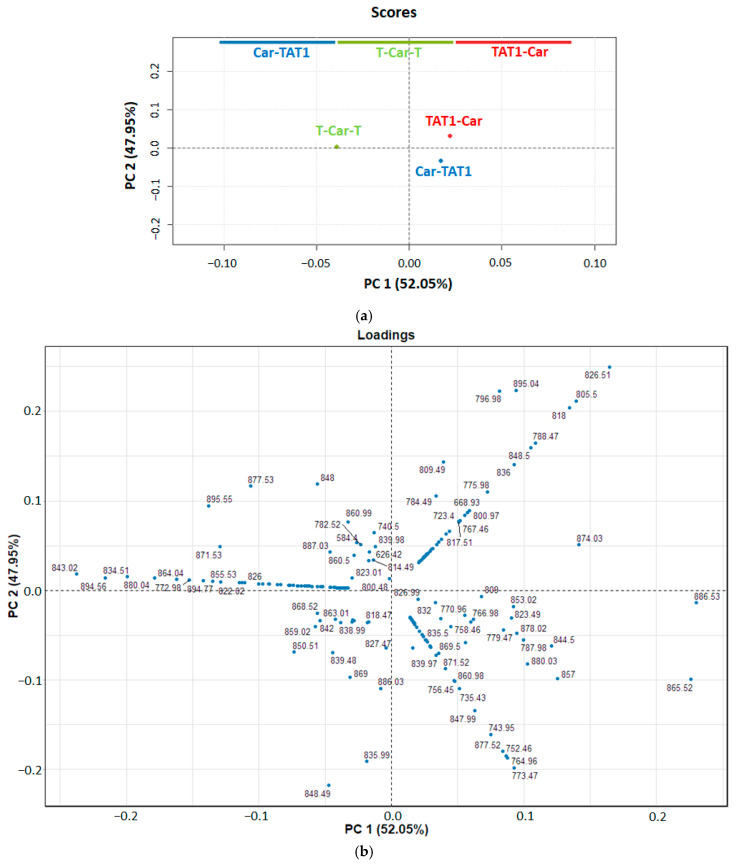
Principal Component Analysis (PCA) performed on the high-resolution MS/MS spectra of T-Car-T, TAT1-Car, and Car-TAT1 in the *m*/*z* range of 100–900. (**a**) Score plot of the two first principal components (the explained variance for PC1 is 52.05% and 47.95% for PC2). (**b**) Loading plot of PC1 vs. PC2. Each point (in blue) shows the loading of a mass peak (*m*/*z*) for PC1 (in *x*-axis) and PC2 (in *y*-axis).

**Figure 5 molecules-30-00853-f005:**
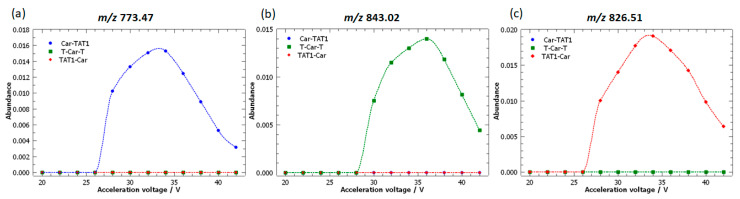
Breakdown curves of the doubly protonated isomeric peptides: Car-TAT1 (blue); TAT1-Car (red); and T-Car-T (green), corresponding to the key fragment ions peak identified as the most significant loadings. The high-resolution MS/MS spectra were acquired at acceleration voltages ranging from 20 V to 42 V.

**Figure 6 molecules-30-00853-f006:**
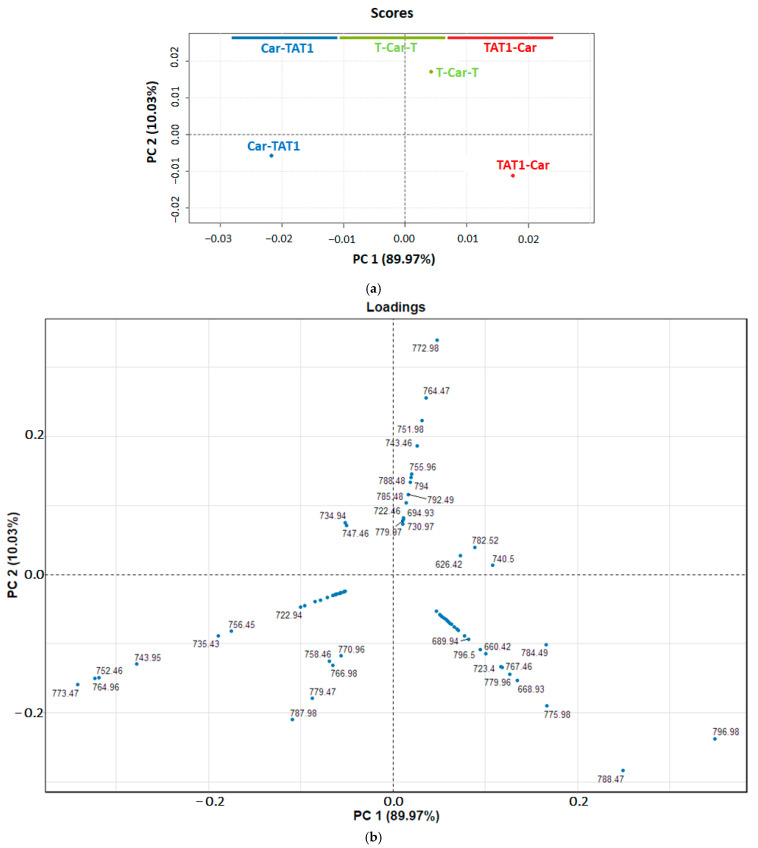
Principal Component Analysis (PCA) performed on the high-resolution MS/MS spectra of T-Car-T, TAT1-Car, and Car-TAT1 in the reduced *m*/*z* range of 600–800. (**a**) Score plot of the first two principal components, with PC1 explaining 89.97% of the variance and PC2 explaining 10.03%. (**b**) Loading plot of PC1 versus PC2. Each blue point represents the loading of a mass peak (*m*/*z*), with PC1 shown on the *x*-axis and PC2 on the *y*-axis.

**Figure 7 molecules-30-00853-f007:**
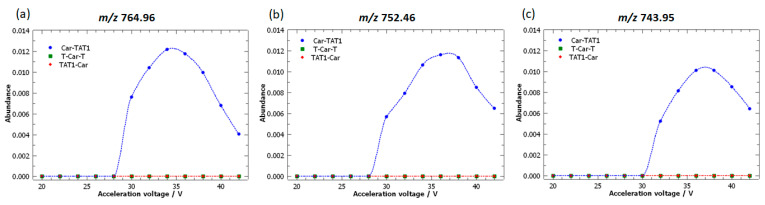
Breakdown curves of the doubly protonated isomeric peptides: Car-TAT1 (blue); TAT1-Car (red); and T-Car-T (green). The curves correspond to fragment ions identified as characteristic of Car-TAT1 based on the loadings plot. High-resolution MS/MS spectra were acquired at acceleration voltages ranging from 20 V to 42 V.

**Figure 8 molecules-30-00853-f008:**
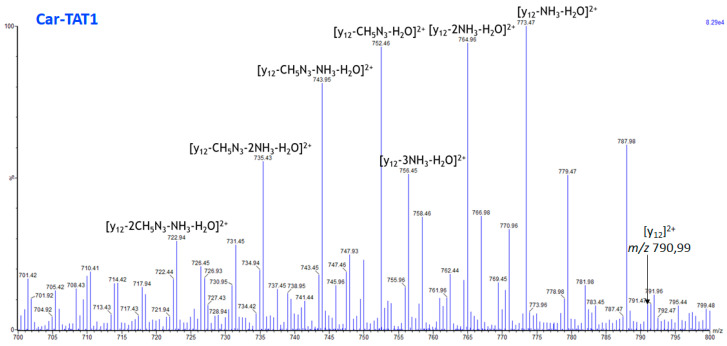
Extract from the high-resolution MS/MS spectrum of the Car-TAT1 peptide, covering the *m*/*z* range from 700 to 800. Ions derived from consecutive neutral losses of [y_12_]^2+^ are highlighted, although [y_12_]^2+^ is hardly detected.

**Figure 9 molecules-30-00853-f009:**
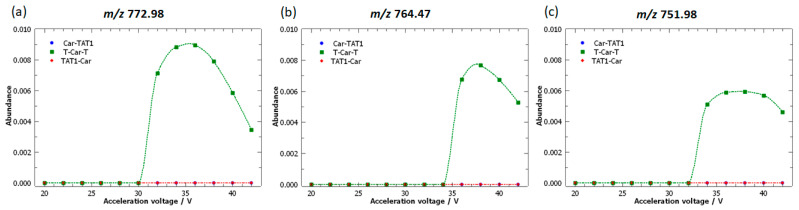
Breakdown curves of the doubly protonated isomeric peptides: Car-TAT1 (blue); TAT1-Car (red); and T-Car-T (green). The curves correspond to fragment ions identified as characteristic of T-Car-T based on the loadings plot. High-resolution MS/MS spectra were acquired at acceleration voltages ranging from 20 V to 42 V.

**Figure 10 molecules-30-00853-f010:**
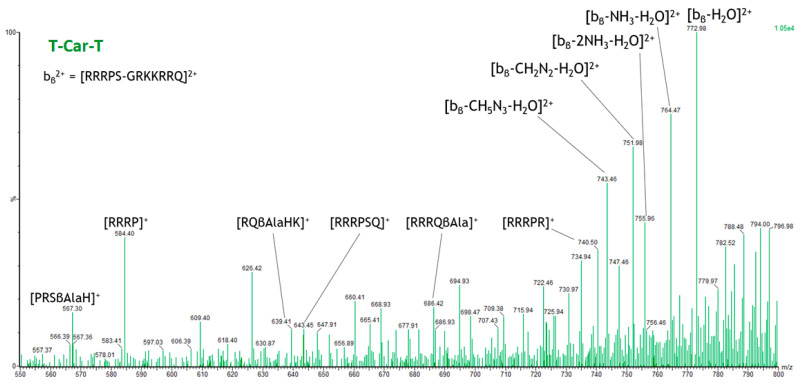
Extract from the high-resolution MS/MS spectrum of the T-Car-T peptide, covering the *m*/*z* range from 550 to 800. Ions derived from scrambling and consecutive neutral losses are highlighted.

**Figure 11 molecules-30-00853-f011:**
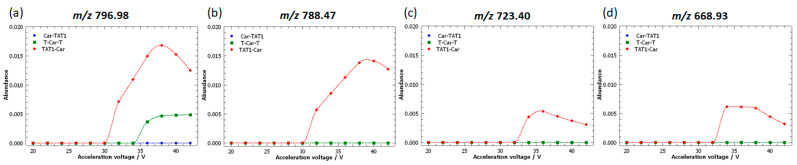
Breakdown curves of the doubly protonated isomeric peptides: Car-TAT1 (blue); TAT1-Car (red); and T-Car-T (green). The curves correspond to fragment ions identified as characteristic of TAT1-Car based on the loadings plot. High-resolution MS/MS spectra were acquired at acceleration voltages ranging from 20 V to 42 V.

**Figure 12 molecules-30-00853-f012:**
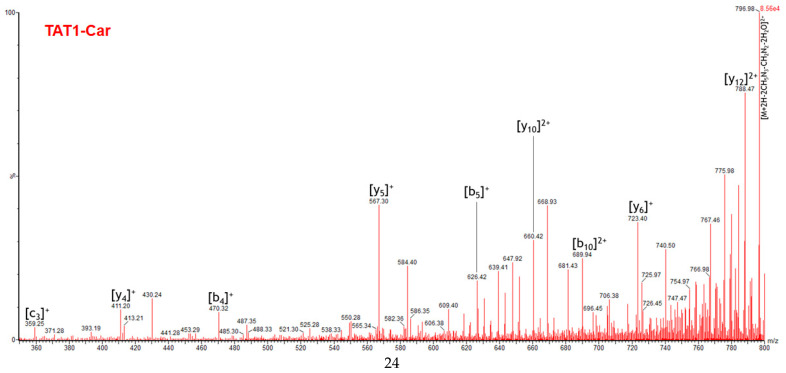
Extract from the high-resolution MS/MS spectrum of the TAT1-Car peptide, covering the *m*/*z* range from 360 to 800. Ions derived from conventional fragmentation are highlighted.

## Data Availability

The data presented in this study are contained within the article.

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
