# Peer review of "Differentiation of Isomeric TAT1-CARNOSINE Peptides by Energy-Resolved Mass Spectrometry and Principal Component Analysis"

_molecules, 2025, doi:10.3390/molecules30040853_

Round 1
Reviewer 1 Report
Comments and Suggestions for Authors
Maroto et al. presented a well structured work demonstrating the potential of PCA as peak identification tool. However, the manuscript should be adjusted before it is considered for publication, a supplementary document should be created.
- Line 127, Figure 1. The observation is, a good peptide dissociation will totally fragment the precursor ion. This allows a variety of fragment ions able to identify all the parts of the peptide. Therefore, I consider the author should increase more the acceleration voltage of its dissociation experiments and present better MS/MS data. Please review if figures 1 and 2 correspond to MS1 data (fullScan) and not mass spectra. If so, please adjust the text accordingly.
- Line140. Figure 3. describe the peptide using the x, y, z, a, b, c fragment types. DOI: 10.4236/jcc.2023.113008 (as described in fig 14). Actually, figure 1 can be sent to a supplementary info section and use figure 3 to describe the peptide fragments.
- Figure 4. this figure is comparing the fragmentation of the precursor across the used acceleration voltages. So, if I understand well in values of 40 the precursor will be almost fully dissociated. But in figure 1 you mentioned an acceleration voltage of 36 and the figure show a large signal for the precursor. Please revise and explain
- In my point of view figures 4 and 5 should be placed into the supplementary material. Even do is an attempt to explain the fragment differences the results are not significant.
- Figure 6. Improve the quality of the figure
- improve the quality of figure 8
- Homogenize the description of the MS/MS of Car-TAT1 (Fig10), T-Car-T (Fig 12), and TAT1-Car (Fig 14).
- The manuscript contains a large number of figures, Ia recommends creating a supplementary document and add the less important plots on it.
The results and discussion sections are lacking references that support the findings and used methodology
Author Response
Please refer to the PDF file for the response to the reviewer's comments.

Reviewer 2 Report
Comments and Suggestions for Authors
After reviewing the manuscript "Differentiation of isomeric TAT1-CARNOSINE peptides by Energy-Resolved Mass Spectrometry and Principal Component Analysis", I have the following comments:
1. Redundant phrases in abstract: Phrases such as "At first glance" and "Interestingly" may not add value and can be omitted for conciseness.
2. The introduction mixes background information about carnosine and TAT1 without clearly transitioning to the rationale for combining them. A more structured approach could help.
3. Figure 1: The figure illustrates the amino acid sequences of carnosine (Car) and TAT1 along with the three isomeric peptides (Car-TAT1, T-Car-T, and TAT1-Car). However, the visual layout of the sequences is plain and could benefit from more graphical enhancement to improve readability and differentiation. Using color coding or highlighting to indicate the specific position of the carnosine residue (e.g., N-terminal, C-terminal, middle) would make the figure more intuitive and visually appealing.
4. Figure 2: The spectra are cluttered, especially in the range of m/z 800–900, where many fragment ions overlap. Without distinguishing features or zoomed-in views, this area is hard to interpret. Also, while the colors correspond to the three peptides, the legend is not sufficiently prominent or clear in explaining this.
5. Figure 5: The figure does not include any explanation or visual aids to highlight subtle differences (e.g., a zoomed-in section or statistical overlays) between the peptides. This could mislead readers into thinking the peptides are indistinguishable.
6. Figure 6: Key features in the plots, such as clusters in the score plot or significant peaks in the loading plot, are not annotated. This makes it difficult for readers to identify which peaks or patterns are responsible for distinguishing the peptides.
Also the figure appears to be of low resolution, making it difficult to discern finer details in the plots, such as data points or subtle variations in the loadings and scores.
7. While the conclusion hints at future directions (e.g., exploring alternative fragmentation techniques like ETD or using ion mobility spectrometry), these ideas are not elaborated or tied back to the findings of the study.
8. Also, the conclusion does not adequately tie the findings back to the broader field of analytical chemistry or peptide analysis, which would help underscore the study's relevance and importance.
Author Response
After reviewing the manuscript "Differentiation of isomeric TAT1-CARNOSINE peptides by Energy-Resolved Mass Spectrometry and Principal Component Analysis", I have the following comments:
- Redundant phrases in abstract: Phrases such as "At first glance" and "Interestingly" may not add value and can be omitted for conciseness.
As suggested, we have omitted the redundant phrases and we have also rewritten the abstract to improve conciseness.
- The introduction mixes background information about carnosine and TAT1 without clearly transitioning to the rationale for combining them. A more structured approach could help.
We have added the following sentences in the introduction to improve transitioning to the rationale for combining them.
The following paragraph was added after Line 54 “… less susceptible to carnosinase degradation.[13,19,20]”:
“In this perspective, one possible strategy is represented by conjugation with peptide chains with high stability and resistant to proteolytic degradation, such as TAT1, an arginine-rich cell-penetrating peptide (CPP) derived from the transactivator of transcription (TAT) protein of HIV-1, which is known for its ability to efficiently traverse cellular membranes.[21–23]”
Then, after line 60, we have added: “Therefore, the combination of…”
- Figure 1: The figure illustrates the amino acid sequences of carnosine (Car) and TAT1 along with the three isomeric peptides (Car-TAT1, T-Car-T, and TAT1-Car). However, the visual layout of the sequences is plain and could benefit from more graphical enhancement to improve readability and differentiation. Using color coding or highlighting to indicate the specific position of the carnosine residue (e.g., N-terminal, C-terminal, middle) would make the figure more intuitive and visually appealing.
We have modified Figure 1 to visually emphasise the specific position of the carnosine residue. The carnosine residue is now highlighted using the same colour scheme applied to each peptide in the manuscript: blue for Car-TAT1, green for T-Car-T, and red for TAT1-Car.
- Figure 2: The spectra are cluttered, especially in the range of m/z 800–900, where many fragment ions overlap. Without distinguishing features or zoomed-in views, this area is hard to interpret. Also, while the colors correspond to the three peptides, the legend is not sufficiently prominent or clear in explaining this.
We have moved Figure 2 to the Supporting Information. The purpose of this figure was to illustrate that the major fragments for all three peptides are located within the 800–900 m/z range. Additionally, we have included zoomed-in views of the three peptides in the Supporting Information to clearly demonstrate that all peptides share the same major fragments within this mass range.
- Figure 5: The figure does not include any explanation or visual aids to highlight subtle differences (e.g., a zoomed-in section or statistical overlays) between the peptides. This could mislead readers into thinking the peptides are indistinguishable.
The purpose of Figures 4 and 5 (the SY plot and the breakdown curve of the major fragment ions) was to demonstrate that the three peptides cannot be differentiated based on energy-resolved mass spectrometry, as the relative intensities are too similar and all the major fragment ions are present in all three peptides. This highlights the need for PCA to identify fragment ions that, although not of higher intensity, are specific to each peptide and could potentially differentiate the three isomeric peptides.
Following the recommendation of the other reviewer, we have revised the discussion and moved Figure 5 (breakdown curves of the major fragment ions) to the Supporting Information. This decision was made because the manuscript already contains many figures, and, as suggested by both you and the other reviewer, these particular figures do not provide valuable information for distinguishing the three peptides.
- Figure 6: Key features in the plots, such as clusters in the score plot or significant peaks in the loading plot, are not annotated. This makes it difficult for readers to identify which peaks or patterns are responsible for distinguishing the peptides.
Also the figure appears to be of low resolution, making it difficult to discern finer details in the plots, such as data points or subtle variations in the loadings and scores.
We have modified the loading plot to clearly identify all peaks and improved the resolution of the figure. For better clarity, we have included both the score plot and the loading plot in the main manuscript. Additionally, the loading plot has been enlarged to facilitate peak identification.
The loading plots (as line plots) have been moved to the Supporting Information (previously Fig. 6c and 8c in the former version of the manuscript). We have also removed the biplot, as its information can be effectively obtained by examining the score and loading plots together.
- While the conclusion hints at future directions (e.g., exploring alternative fragmentation techniques like ETD or using ion mobility spectrometry), these ideas are not elaborated or tied back to the findings of the study.
The last paragraph of the conclusion was modified to include more detailed suggestions with clearer connections to our findings (lines 531-540). Five references were also added (references 64-69).
“Additionally, alternative MS fragmentation techniques, such as Electron-based techniques (ExD), could be employed, as the TAT1 peptides easily generate multiply protonated signals due to their arginine-rich structures. This would help prevent arginine side chain losses, which lack structural information, and instead promote more backbone fragmentations that are more closely related to the peptide's structure which is suitable for the studied sequences. [64–67] Finally, ion-mobility mass spectrometry could be investigated to evaluate its ability to distinguish between the three isomeric peptides based on their gas-phase conformations, as the position of the carnosine dipeptide can induce different structural arrangements, leading to variations in their Collision Cross Section (CCS) and thus measured drift times, thereby enabling differentiation. [68,69]”
- Also, the conclusion does not adequately tie the findings back to the broader field of analytical chemistry or peptide analysis, which would help underscore the study's relevance and importance.
We have added the following paragraphs to the conclusions to emphasise the significance of this study in the broader field of peptide analysis and proteomics. Additionally, six references (59, 70–74) have been included to support and complete the findings :
“It may be worth highlighting that classical peptide sequencing methods using CID activation, employed in most MS/MS instruments and based on the mobile proton model, typically lead to backbone fragmentations through amide bond cleavage.[70] Ideally, this produces comprehensive and uniform fragmentation patterns, generating a broad series of complementary b and y ions. Machine learning based classification methods have been applied in this context to assign their b and y nature, improving the accuracy and efficiency of de novo sequencing.[59,71,72] However, in the case of the studied peptides, charge sequestration occurs due to the presence of many arginine residues, which is the most basic amino acid in the gas phase. As a result, fragmentation deviates from the ideal scenario, leading to multiple uninformative neutral losses and parasitic fragmentation pathways, such as the formation of internal fragments and scrambled sequences. This significantly complicates spectral assignment. [73,74] Moreover, no reference spectral databases are available for comparison, as these compounds are not true peptides due to the presence of a beta amino acid. Consequently, despite extensive knowledge and methodologies for peptide sequencing using MS/MS, such complex and atypical cases require dedicated investigations. This highlights the need for complementary and orthogonal approaches, such as the PCA, for identifying non conventional fragments, as demonstrated in the present study.”

Round 2
Reviewer 1 Report
Comments and Suggestions for Authors
The authors have addressed my initial concerns, now I agree with the publication of the manuscript
Reviewer 2 Report
Comments and Suggestions for Authors
The revisions have greatly improved the manuscript’s clarity and flow. The abstract is more concise, and the introduction transitions smoothly, providing a clear connection to the study’s purpose. The updated Figure 1, with its color coding, and the relocation of Figures 2 and 5 to the Supporting Information help streamline the content, making it easier to follow. The enhanced resolution and annotations in Figure 6 also make the PCA results much clearer and more accessible. The expanded conclusion does a great job of tying the findings to the bigger picture of peptide analysis and suggesting meaningful future directions. Overall, the key concerns have been well addressed, and the manuscript is now much stronger. Just a small note, ensure all supporting materials are properly referenced in the main text.